# Metabolite Profiling of Conifer Needles: Tracing Pollution and Climate Effects

**DOI:** 10.3390/ijms241914986

**Published:** 2023-10-08

**Authors:** Marius Gheorghe Miricioiu, Roxana Elena Ionete, Svetlana Simova, Dessislava Gerginova, Oana Romina Botoran

**Affiliations:** 1ICSI Analytics Group, National Research and Development Institute of Cryogenic and Isotopic Technologies–ICSI Rm. Vâlcea, 4 Uzinei Street, 240050 Râmnicu Vâlcea, Romania; marius.miricioiu@icsi.ro (M.G.M.); roxana.ionete@icsi.ro (R.E.I.); 2Bulgarian NMR Centre, Institute of Organic Chemistry with Centre of Phytochemistry, Bulgarian Academy of Sciences, “Acad G. Bonchev” Street, Bl. 9, 1113 Sofia, Bulgaria; svetlana.simova@orgchm.bas.bg (S.S.); dessislava.gerginova@orgchm.bas.bg (D.G.)

**Keywords:** metabolomics, bioindicators, air pollution, NMR, multivariate statistical analysis

## Abstract

In the face of escalating environmental challenges, understanding the intricate relationship between plant metabolites, pollution stress, and climatic conditions is of paramount importance. This study aimed to conduct a comprehensive analysis of metabolic variations generated through ^1^H and ^13^C NMR measurements in evergreen needles collected from different regions with varying pollution levels. Multivariate analyses were employed to identify specific metabolites responsive to pollution stress and climatic factors. Air pollution indicators were assessed through ANOVA and Pearson correlation analyses. Our results revealed significant metabolic changes attributed to geographical origin, establishing these conifer species as potential indicators for both air pollution and climatic conditions. High levels of air pollution correlated with increased glucose and decreased levels of formic acid and choline. Principal component analysis (PCA) unveiled a clear species separation, largely influenced by succinic acid and threonine. Discriminant analysis (DA) confirmed these findings, highlighting the positive correlation of glucose with pollution grade. Beyond pollution assessment, these metabolic variations could have ecological implications, impacting interactions and ecological functions. Our study underscores the dynamic interplay between conifer metabolism, environmental stressors, and ecological systems. These findings not only advance environmental monitoring practices but also pave the way for holistic research encompassing ecological and physiological dimensions, shedding light on the multifaceted roles of metabolites in conifer responses to environmental challenges.

## 1. Introduction

Air pollution is a delicate and concerning problem for our present and future society mainly due to rapid industrialization and urbanization. These factors directly affect human health and the entire ecosystem with prolonged exposure. The first impact of pollutants on plants is visual, resembles the effects of drought stress, and is expressed in slow growth and surface appearance [1,2,3]. The mechanism of pollutant accumulation in plants is different and consists in particulate retention, stomatal gas exchange, and surface ion exchange [4]. The absorption capacity or rate of contaminants depends on the plant species and can be strongly influenced by physical and chemical factors, such as solubility, hydrophobicity, vapor pressure, particle size, metal oxidation state, and environmental temperature and humidity [1,4,5]. Therefore, various studies have explored the potential use of trees as sensors for air monitoring, exploiting their large surface area that allows them to capture certain pollutants through their bark, leaves, or needles [2,3,5,6,7]. Fungi, lichens, and mosses have also attracted attention in pollution assessment, but they present a serious drawback due to the difficult differentiation of similar species [5,6]. Conifers possess a particular advantage over deciduous trees because of their evergreen nature, which allows them to accumulate and store airborne pollutants for several years. This accumulation history makes it possible to establish long-term air pollution levels in certain areas by separating needles that vary from 1 to 3 years on the same branch [4,7]. Persistent organic pollutants have poor water solubility, and the ability of lipidic tissues to accumulate low vapor pressure allows the wax layer to retain pollutants from the atmosphere for an extended period [7]. Additionally, due to their lipophilic characteristics, hydrocarbons accumulate in plants, but the most common studies involving isotopes, metals, ammonia, and nitrogen have focused on inorganic air pollutants [6]. Beside physiological changes, plants can adapt to climatic conditions and pollution changes through structural and morphological modifications, using internal resources that lead to stress reduction [3,8]. These adaptive responses often involve changes in leaf surface properties, root architecture, and secondary metabolite production. For example, some plants may develop thicker cuticles or altered stomatal density to minimize pollutant entry, while others may produce specialized metabolites that aid pollutant detoxification [9]. Understanding these diverse adaptation mechanisms is critical for assessing the resilience of plant species to ongoing environmental challenges.

Therefore, a better understanding of metabolite variations or physiological adjustments as adverse responses to pollution stress in conifers from different areas (e.g., industrial, urban, rural) can provide important information about pollutants at the environmental level. These plants have great potential to be used in the future as biomonitors due to their ability to assimilate contaminants over a long period of time, reflecting environmental conditions and stress phenomena. The metabolites that can be identified in plants differ in terms of structure, compound families, and concentrations. A comprehensive analysis of these metabolites can offer insight into the specific pollutants present in the environment and their potential impact on ecosystems [10]. Integrating data obtained from plant biomonitors with other environmental monitoring techniques can improve our understanding of pollution sources and pathways, leading to more effective strategies for pollution control and environmental protection.

In this way, environmental conditions and plant physiology can arise through the formation of certain compounds or a change in their concentration that already exist. Stable isotopes analysis has been widely used to determine climatic conditions and geographic origins [10] via the chemical, physical, and biological processes that affect isotope fractionation. In addition, researchers have used methods such as DNA barcoding and genetic sequencing to study the genetic diversity and population structure of plants in response to environmental stressors [11,12]. These molecular approaches offer valuable insights into how plant species adapt and evolve under different pollution scenarios. A solvent extraction method followed by HPLC with fluorescence detection has been applied to determine the content of polycyclic aromatic hydrocarbons (PAHs) in leaves, needles, and grass, demonstrating their ability to accumulate the contaminants in tissues and thereby reflecting the impact of anthropogenic activities [13,14]. Furthermore, multivariate statistical analysis based on matrix-assisted laser desorption-mass spectrometry (MALDI-MS), surface desorption atmospheric pressure chemical ionization mass spectrometry (DAPCI-MS), gas chromatography-mass spectrometry (GC-MS), and data from nuclear magnetic resonance (NMR) have been used in several studies to compare and identify differences in chemical or metabolite compositions. These specific differences may be related to origin, soil quality, growth conditions, or other parameters such as the age, sex, and mating status of the plants [15,16,17,18,19]. Therefore, the use of conifer needles for air pollution monitoring can be a cost-effective and reliable solution, especially in hard-to-reach areas where energy connections or dedicated monitoring analyzers are not practicable. Additionally, the response of conifers to air contaminants may provide advanced warning signs of rising air pollution levels. Furthermore, the integration of advanced remote-sensing technologies, such as hyperspectral imaging and drone-based monitoring, can complement plant-based biomonitors for comprehensive air pollution assessments over larger geographical areas [20,21]. Moreover, by combining plant biomonitors with real-time data from atmospheric sensors and weather stations, environmental authorities can develop more effective strategies for pollution control and timely response to pollution events [22]. Harnessing the potential of these innovative techniques will further improve our understanding of air quality dynamics and contribute to the protection of both human health and ecosystems.

Most of the related studies have focused on determining different contaminant levels in plant tissues, but the relationship between plant metabolites and the impact of pollution remains poorly understood. Therefore, the aim of this study was to strengthen and complement the existing assumptions found in the literature by identifying correlations between metabolites in spruce or fir trees and pollution levels in specific areas where they grow using ^1^H NMR profiling and multivariate statistical analysis.

## 2. Results and Discussion

^1^H 1D and ^1^H–^13^C 2D HSQC (heteronuclear single quantum coherence) NMR spectra were acquired to identify the metabolites in needle-extract samples for further studies regarding their correlations with pollution level. Figure 1 shows a typical 600 MHz proton NMR spectrum with water suppression and the corresponding ^1^H–^13^C 2D HSQC NMR spectrum. We were able to identify several metabolites in different chemical shift regions. In the high field or aliphatic region (δ 0.5–3.0 ppm), we identified signals corresponding to threonine, alanine, GABA (γ-aminobutyric acid), and acetic and succinic acids. In the carbohydrate region (δ 3.0–5.5 ppm), signals of α-glucose, β-glucose, fructose, and sucrose are prominent. Additionally, the low field or aromatic region (δ 5.5–9.0 ppm) exhibits signals indicative of shikimic and formic acids. The identified metabolites and their corresponding chemical shifts and multiplicities are summarized in Table 1. The presence of amino acids in needle extracts was expected given their important contribution to protein biosynthesis and their essential roles in tree growth and development, intracellular pH regulation, metabolic energy generation, and protection from abiotic/biotic stress [23,24,25]. Sucrose, α- and β-glucose, and fructose were identified as the dominant sugars. The detection of these sugars within the conifer needles serves as a biochemical indicator affirming the developmental maturity of the examined samples. These sugars hold crucial significance as primary substrates involved in the process of photosynthesis, wherein they contribute to the synthesis of energy-rich molecules and carbon assimilation. This carbon and energy, in turn, play essential roles in sustaining various physiological processes throughout the coniferous organism, including the metabolic demands of root systems and the vigorous growth of nascent needles. In the same region, choline was also identified at a chemical shift of 3.21 ppm, which has an important role in maintaining the structural integrity of plants and is involved in various metabolic processes [23]. The signal associated with formic acid is found at a chemical shift of 8.48 ppm. Formic acid, along with acetic acid, is produced in needles during plant metabolism as a result of the decarboxylation of glycolic acid during photorespiration and oxidation of formaldehyde in needles or leaves [26,27,28]. There are many reports on formic acid emissions from forests to the atmosphere, especially during the growing season of trees. These emissions are of ecological significance because formic acid can act as a signaling compound between plants, participating in plant–plant interactions and defense responses against herbivores [12,26,29,30]. Similarly, acetic acid is formed after the hydrolysis of acetyl-CoA [31] and decarboxylation of acetaldehyde in leaves or needles [32]. As in the case of formic acid, acetic acid is released in a gaseous form into the atmosphere by leaves or needles [26]. Acetic acid emissions from vegetation affect atmospheric chemistry and contribute to the formation of secondary organic aerosols that influence air quality and cloud formation. Additionally, acetic acid may play a role in plant signaling and defense against pathogens.

Identification of the listed metabolites in spruce and fir needle extracts provides valuable insights into the metabolic processes and environmental stress responses in these conifers. The role of these compounds as potential biomarkers of pollution levels and their involvement in plant interactions and atmospheric processes highlights the importance of such studies in understanding the ecological consequences of air pollution on forest ecosystems. In order to observe the variations of metabolic signals and to establish their correlation with pollution, the obtained data were further submitted to statistical analysis.

One-way ANOVA analyses of variance (*p* < 0.05) with pairwise post hoc comparison using Tukey’s test were employed to determine the significance of differences in the metabolites present in needle extracts collected from four different regions with varying pollution levels. According to Table 2, the metabolites with the highest concentration in all regions were, in descending order, shikimic acid, β-glucose, succinic acid, α-glucose, and fructose. Among them, the major component, shikimic acid, accounted for between 26% and 39% of the total metabolite content in needle extracts. Several metabolic changes have been observed in response to pollution stress. The ANOVA results indicated an increasing trend of β-glucose and α-glucose levels in needle extracts due to pollution, as well as a slow decrease in formic acid and choline levels, which was more visible between region 1 (unpolluted) and region 4 (polluted). These findings are consistent with previous research highlighting the role of glucose as a stress-responsive metabolite, potentially related to plant adaptive mechanisms under pollution-induced stress [33,34]. A decrease in formic acid and choline, compounds involved in structural integrity and metabolic processes [35], may indicate a disturbance in these vital functions under the influence of pollution.

In the first three regions, there was a trend for increased threonine levels with higher pollution levels. Threonine is known to be involved in nitrogen metabolism and stress responses, acting as a precursor for multiple downstream pathways, including the biosynthesis of secondary metabolites involved in defense against stressors [36]. Furthermore, polluted regions showed a gross difference from the less polluted regions based on succinic and shikimic acid values. Elevated levels of succinic acid, a key component of the tricarboxylic acid cycle, may indicate changes in energy metabolism and cellular respiration under pollution stress. Shikimic acid, a precursor for the synthesis of aromatic amino acids and secondary metabolites, may potentially indicate shifts in the plant’s allocation of resources toward defense responses against pollutants. While these trends were observed in our data, they were not generally related with pollution, and it is important to note that further research is needed in order to establish the significance and mechanisms behind these observations. However, variations in the amounts of GABA, succinic acid, and alanine showed no clear differences in response to pollution, making it challenging to establish any correlations.

To facilitate clear visualization, all data obtained for the aliphatic, carbohydrate, and aromatic regions of the ^1^H NMR spectrum were subjected to Pearson analysis and represented as a heatmap in Figure 2. This heatmap was used to assess variation in metabolite concentrations among needle extract samples from the collected areas. As expected, the relationship between precipitation and annual mean minimum and maximum temperature was the strongest. Higher altitudes typically experience more intense precipitation and lower temperatures. Regarding metabolites, a significant and negative correlation was observed between formic acid and climatic conditions, with a positive correlation with elevation. Choline also exhibited a similar pattern, highlighting its sensitivity to both climatic and elevation factors. These findings are consistent with previous research that noted the interplay between environmental conditions and metabolite levels, suggesting that formic acid and choline may serve as potential indicators of the combined effects of climate and altitude on conifer metabolism [37]. It should be noted that there was a strong positive correlation between α- and β-glucose and climatic conditions. Glucose showed accumulation in needles and a negative correlation with altitude, possibly reflecting its role as an energy source in response to favorable climatic conditions. The relationship between glucose accumulation and climate is consistent with an adaptive strategy of trees to optimize energy storage during periods of favorable growth conditions. In contrast, weaker or absent correlations were observed for metabolites belonging to the amino acid class, suggesting that their levels may be less affected by climatic or elevation gradients. Additionally, the lack of a clear correlation for sucrose and fructose, despite other carbohydrates showing different patterns, indicates the complexity of the carbohydrate metabolism in conifers and the potential involvement of specific regulatory mechanisms in response to environmental cues.

Principal component analysis (PCA) was used to manage the extensive dataset and reduce its dimensionality. This analytical technique helped to identify spectral variations in regions subject to different degrees of pollution. The resulting PCA model was constructed using five components, including F1, F2, F3, F4, and F5, which contributed 30.21%, 18.47%, 13.27%, 9.91%, and 8.83%, respectively, to the total variance. The score plot depicted in Figure 3 was derived from the first two principal components, collectively explaining 48.67% of the total variance. Notably, a visual assessment of the score plot revealed a discernible trend corresponding to the pollution levels. The distribution pattern observed in the score plot highlights the potential of PCA to be a powerful tool for distinguishing pollution-related metabolic variations. Separating samples of different pollution levels along the F1 axis suggests a gradation of the pollution impact. Moreover, the clear separation of samples from region 1 with negative F1 values from those of regions 2, 3, and 4 with positive F1 values underscores the robustness of the PCA model in capturing underlying trends. PCA-derived insights provide valuable preliminary evidence of the interplay between metabolite profiles and pollution levels, offering a foundation for subsequent in-depth analyses. Furthermore, altitude emerges as a significant variable influencing the separation of region 1 from the other regions: at higher altitudes, pollution sources are absent. The main component responsible for this separation, F1, exhibited correlations with shikimic acid, choline, and formic acid. It should be noted that the variation of these variables appeared to depend on the pollution level, suggesting a potential link between the elevated regions and individual metabolic responses to pollution stress. The second major component, F2, accounting for 18.47% of the total variance, demonstrated correlations with GABA, sucrose, fructose, α- and β-glucose, threonine, and succinic acid. This intricate network of correlations underscores the multifaceted nature of metabolic adaptations In conifers in response to environmental pressures. Interestingly, PCA analysis also detected discernible differences in metabolite composition based on tree species, with all fir samples aligning with negative values on the F2 axis. The separation of species is due, in particular, to succinic acid and threonine, both of which exhibited positive values along the F2 and F1 axes, respectively. This suggests that the observed metabolic distinctions between spruce and fir may be due, at least partially, to succinic acid and threonine levels. These findings shed light on the potential biochemical underpinnings of species-specific responses to pollution and environmental stress.

Intriguingly, as the PCA model captures the cumulative effects of multiple variables, the combined impact of elevation and species on conifer metabolism becomes more evident. By dissecting the complex interplay between metabolite profiles, altitude, and species type, the PCA approach presents a valuable framework for unraveling the intricate mechanisms by which environmental factors shape plant metabolism. These insights not only contribute to our fundamental understanding of plant responses to changing environments but also offer a basis for developing targeted strategies for the conservation and management of coniferous ecosystems.

For a more detailed and comprehensive examination, aiming to validate the insights gained from the explorative PCA analysis and to discover new correlations, the same dataset was subjected to discriminant analysis (DA). This approach involved classifying the samples based on varying pollution levels, thereby offering a finer distinction between them. As it can be seen from Figure 4, the first and second discriminant functions explained 98.26% and 1.25% of the total variance, respectively. The DA model encompassing all metabolites effectively categorized the samples based on their respective regions of origin. This classification underscored the significant quantitative changes in needle metabolites attributed to pollution levels, clearly elucidated by the first discriminant function (F1). Metabolic adaptations to pollution stress exhibited diverse responses, reflecting the intricate strategies used by conifers to cope with environmental challenges. Clear trends emerged, particularly in the increasing statistical distances observed between the least polluted regions (mountainous)—denoted as region 1—and those regions exposed to varying pollution levels. The separation between region 1 and the more polluted regions (regions 2, 3, and 4) highlights the capacity of DA to capture even subtle variations in metabolic profiles and their relationship to pollution gradients. Metabolites exhibiting notable variations in response to F1 further emphasized these trends. Formic acid and choline, characterized by negative coefficients, decreased in concentration with increasing pollution levels, indicating their potential as markers of pollution impact. In contrast, α- and β-glucose, with positive coefficients, exhibited increased concentrations in response to the pollutant presence, confirming the trends observed in the PCA classification. This reinforces the importance of these metabolites as indicators of pollution-induced metabolic shifts.

A subtle separation between region 2, considered relatively unpolluted, and the contaminated regions 3 and 4 was also discernible based on F1. This observation suggests a progressive shift in metabolic profiles as pollution levels increase, further substantiating the utility of DA in elucidating nuanced responses to pollution stress. Additionally, F2 contributed significantly to the discrimination between region 3 and region 4. High-impact metabolites driving this separation included fructose, sucrose, and succinic acid, with each being associated with positive coefficients. Conversely, shikimic acid, alanine, and GABA exerted a negative influence. These findings unveil potential metabolic signatures that differentiate regions exposed to varying pollution intensities, thereby offering insights into the specific compounds implicated in conifer response to pollution.

The accuracy of the confusion matrix achieved in distinguishing samples from unpolluted to polluted zones demonstrated strong performance: 100% accuracy for region 1, 84.62% for region 2, 81.82% for region 3, and 75% for region 4. Together, these results underscore the robustness of the DA analysis in successfully capturing pollution-induced metabolic variations and differentiating conifer samples at varying pollution levels.

To summarize, fir and spruce needles collected from different areas with varying pollution levels were analyzed to highlight metabolic changes within the trees. The results clearly demonstrate that the metabolite composition of needle extracts exhibited significant variation based on their geographical origin, reaffirming the potential of spruce and fir as sensitive indicators for monitoring air pollution and climate conditions. These tree species could effectively serve as passive air samplers, capturing and “recording” pollution levels over time. Multivariate statistical analyses were employed to identify the specific metabolites linked to pollution stress and climatic conditions. Exposure of spruce or fir to a high level of air pollution resulted in an increase in glucose concentration and was accompanied by a decrease in formic acid and choline levels according to ANOVA analysis. Pearson correlation coefficients revealed both negative and positive correlations of formic acid with climatic conditions and altitude, respectively. Conversely, α- and β-glucose exhibited contrasting correlations. PCA enabled the differentiation between the two species, mainly driven by variations in succinic acid and threonine. This analysis found a negative association between shikimic acid, choline, and formic acid levels with pollution intensity. DA analysis substantiated these findings and revealed a positive correlation between α- and β-glucose, and the pollution grade was consistent with the ANOVA outcomes. These results suggest a complex interplay between pollution stress and tree metabolism, offering insights into the biochemical responses of spruce and fir to environmental challenges. The implications of these findings extend beyond pollution assessment. The observed variation in metabolite profiles may also have implications for ecological interactions between these conifer species and other organisms in their environment. The potential roles of these metabolites in defense mechanisms, nutrient cycling, and carbon allocation warrant further investigation, opening avenues for holistic studies that encompass both ecological and physiological dimensions.

## 3. Materials and Methods

### 3.1. Sample Identification, Collection and Preparation

^1^H and ^13^C NMR spectra and various 2D spectra (such as JRES, COSY, TOCSY, and HSQC) were obtained to comprehensively evaluate the correlations of spruce and fir metabolomics with air pollutants. The instrument used for sample characterization was a Bruker Avance NEO 600 spectrometer (Biospin GmbH, Rheinstetten, Germany) equipped with a nitrogen-cooled Prodigy cryoprobe. In addition to the methodologies used to identify metabolites, the statistical data exploration, including PCA and DA, was essential to observing the contribution of each metabolite as a possible stress marker of pollution and to establishing the confidence level of our approach. To meet these requirements, needle samples were collected (using clean rubber gloves for each sample to avoid cross-contamination) from spruces and firs growing in areas at different altitudes and exposed to different pollution levels. Conifers were selected at the same age, approximately 40 years, and were sampled from the same southeast-oriented section at 1.5 m height in the first part of the day. After collection, the samples were stored in polyethylene bags, coded (according to Table 1), and brought to the laboratory for chemical extraction. Table 3 shows information about the selected research sites and their classification according to the pollution level (ranging from low to high), which strongly depends on the type of zone (rural, mountain park, spa, urban, and industrial). The protocol implemented for the extraction procedure was based on different literature studies [38,39,40,41]. Twig pieces were placed in liquid nitrogen, and needles were removed by agitation. The samples were then ground with a ball mill (Pulverisette 6, Fritsch, Germany) and subjected to a 12 h lyophilization process. An amount of 0.05 g of fine powder from each sample was introduced into a 2 mL Eppendorf tube, to which 750 µL of methanol-D4 (99.80% D, VWR Chemicals, Lutterworth, UK) and 375 µL of potassium phosphate buffer solution were added for the extraction and after was filled with deuterium oxide (99.9%). The buffer solution also contained 0.1% 3-(trimethylsilyl) propionic-2, 2, 3, 3-d4 acid sodium salt (99.9% Sigma Aldrich, St. Louis, MO, USA) as a reference for the ^1^H NMR spectra at 0.00 ppm. The mixtures were kept in the vials for 1 h and then sonicated for 15 min without a temperature program. Phase separation was achieved using centrifugation for 20 min, and the supernatant was transferred to a 5 mm NMR tube for analysis after filtration with a PTFE membrane (0.45 µm, Millipore, Burlington, Massachusetts, USA). Sampling for the pollution stress response study was performed four times, and special attention was paid to the uniformity to avoid interference in the metabolomic NMR analysis.

### 3.2. 1D and 2D NMR Spectroscopy

The measurements were performed at a temperature of 300.0 ± 0.1 K. ^1^H NMR spectra were acquired with water signal suppression. The following parameters were applied: *noesygppr1d*, 128 scans, 16 dummy scans, 4 s acquisition time, 2 s relaxation delay, 64K FID size data points, and 13.66 ppm spectral width. For ^13^C NMR experiments (*zgdc*), the following parameters were employed: 30˚ pulse, 4K scans, 16 dummy scans, 1.05 s relaxation delay, 32K data points, and 236.63 ppm spectral width. All spectra were phased, baseline-corrected, and referenced to the TSP signal at 0 ppm for ^1^H NMR spectra and the methanol signal at 49.15 ppm for ^13^C NMR spectra. The chemical shifts of the signals, along with their multiplicity allowing the identification of metabolites, were revealed in the NMR spectra using one-dimensional (1D) and two-dimensional (2D) experiments, including homonuclear ^1^H-^1^H correlation spectroscopy (COSY), total correlation spectroscopy (TOCSY), and J-resolved (JRES) and heteronuclear ^1^H-^13^C single-quantum coherence (HSQC) spectral analysis. The COSY (*cosygpmfqfpr*) spectral width was set to 11.90 ppm, with a 2 s relaxation delay, 4K × 256 increments, 16 dummy scans, and 2 scans for data acquisition. The parameters used in TOCSY experiments (*dipsi2esgpphzs*) were spectral width 11.90 ppm, relaxation delay 2 s, 2K × 256 increments, 2 scans, and 16 dummy scans. JRES (*jresgpprqf*) spectra were obtained with a spectral width of 11.90 ppm for F1 and 66.00 Hz for F2, a relaxation delay of 2 s, 8K × 64 increments, 16 dummy scans, and 4 scans. For the HSQC (*hsqcedetgpsisp2.2*) investigations, the parameters used were achieved with a relaxation delay of 1.5 s, 2K/200 data points in the direct/indirect dimension, 32 dummy scans, 8 scans, and a spectral width of 11.90 ppm for proton and 180 ppm for carbon dimensions. 

### 3.3. Metabolite Identification and Quantification

After the NMR spectra were collected and manual phasing and baseline correction were performed, the metabolites were unambiguously identified via 1D and 2D NMR spectra. Furthermore, to ensure reliable identification of metabolites, various literature studies were consulted to compare the spectra with those in previously published research [19,23,42,43]. Metabolite intensity values were subsequently registered in a Microsoft Excel spreadsheet for multivariate statistical analysis, and each one was reported as percentage of the total signals that were taken into consideration. 

### 3.4. Statistical Analysis

The potential correlation of air pollution with some metabolite variations in spruce or fir needle extracts from NMR data was evaluated by a combination of established analytical tools, including analysis of variance (ANOVA), principal components analysis (PCA), and discriminant analysis (DA). The statistical analysis helped overcome the challenges posed by multiple ^1^H NMR spectra signals, allowing for the contribution of the molecules responsible for key differences to be highlighted. 

First, ANOVA analysis was performed to examine the trends of metabolites in four different regions with respect to pollution. Additionally, Pearson correlation was applied to identify the correlation coefficient and to strengthen the similarities in terms of metabolite presence. To further identify components that can be used as pollution markers, we conducted PCA followed by DA to examine the data more closely using Addinsoft XLSTAT software version 2014.5.03 (Addinsoft Inc., New York, NY, USA). These statistical approaches helped to understand the relationships between metabolites and pollution levels and to identify potential pollution biomarkers and their significance in the overall metabolic profile.

## 4. Conclusions

The present study highlights the dynamic nature of conifer metabolism in response to pollution stress and climatic conditions. Using advanced analytical techniques and statistical analyses, a comprehensive understanding of the intricate relationships between metabolites, environment, and species behavior has been elucidated. These findings not only contribute to environmental monitoring practices but also stimulate broader research exploring the ecological and functional consequences of these metabolic adaptations in conifers.

## Figures and Tables

**Figure 1 ijms-24-14986-f001:**
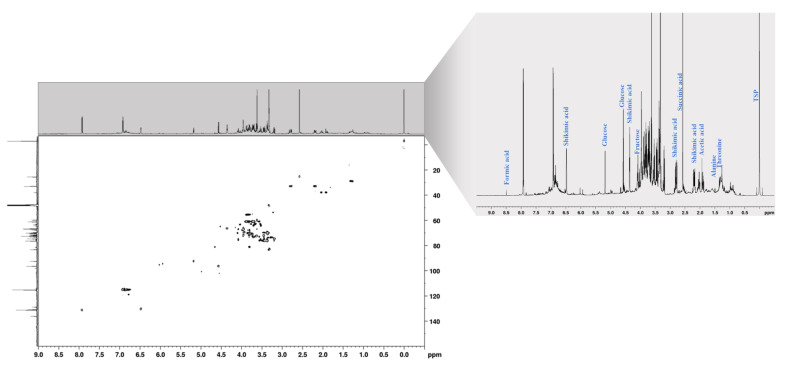
^1^H–^13^C 2D HSQC NMR spectrum with a partially assigned 600 MHz ^1^H 1D NMR spectrum of needle extract (300.0 ± 0.1 K, methanol-D_4_, pH 3, 20 mM TSP).

**Figure 2 ijms-24-14986-f002:**
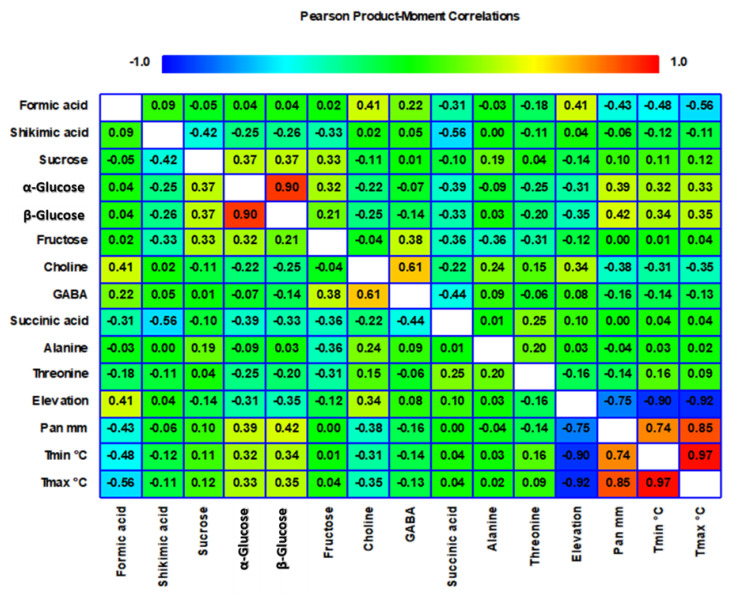
Pearson correlation heatmap revealing metabolite concentration variations in needle extracts across regions.

**Figure 3 ijms-24-14986-f003:**
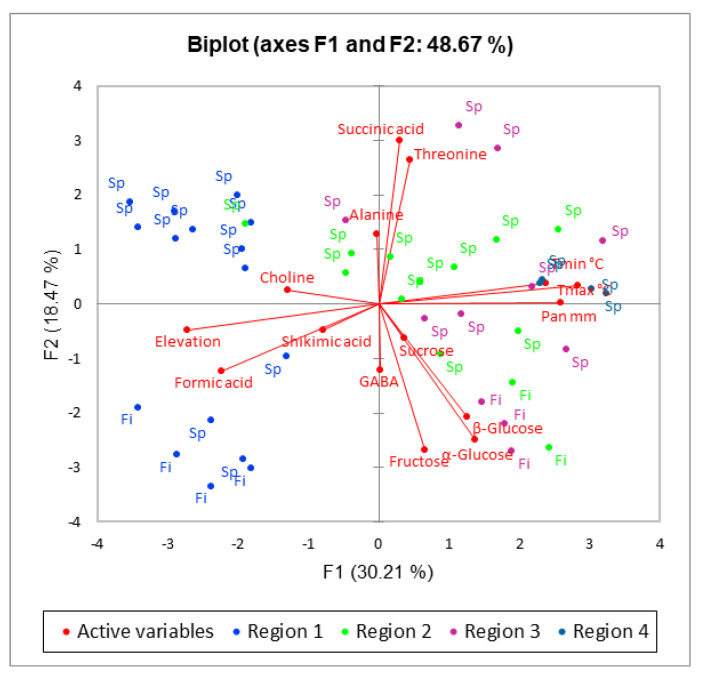
PCA score plot of the samples collected from four regions and the correlation between signals and metabolites responsible for pollution differentiation.

**Figure 4 ijms-24-14986-f004:**
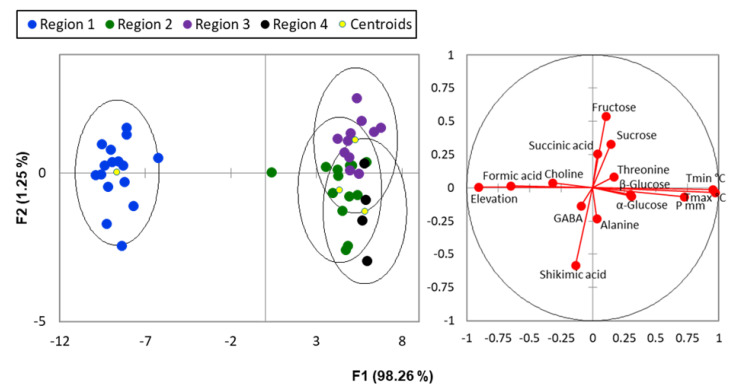
Discriminant analysis F1/F2 score plot showing the separation between regions in terms of their pollution level and the main metabolites responsible for it.

**Table 1 ijms-24-14986-t001:** Assigned ^1^H and ^13^C NMR chemical shifts, multiplicities, and J coupling constants of metabolites in needle extracts.

Compound	^1^H δ (ppm) *	Multiplicity **	J (Hz)	Group	^13^C δ (ppm) *
Amino acids					
Threonine	1.33	d	7.2	CH_3_	19.8
	4.04	m	-	CH	n.d. ***
Alanine	1.49	d	7.2	CH_3_	17.2
	3.70	m	-	CH	n.d.
GABA	2.32	t	7.1	CH_2_(NH_2_)	35.5
	1.91	m	-	CH_2_	25.2
	3.00	t	7.3	CH_2_(COOH)	40.5
Carbohydrates					
Fructose	3.92			CH	69.5
				C	98.0
				C	101.7
				C	104.4
β-glucose	4.56	d	8.0	CH(O)	97.6
	3.19	dd	8.0; 9.3	CH	75.8
	3.43	t	9.3	CH	77.5
	3.31	dd	7.9; 9.3	CH	71.2
	3.37	m	-	CH	77.6
	3.87, 3.69	m	-	CH_2_	62.4
α-glucose	5.17	d	3.9	CH(O)	93.6
	3.45	dd	3.9; 9.5	CH	73.3
	3.62	t	9.5	CH	74.4
	3.34	m	-	CH	71.3
	3.82	m	-	CH	72.9
	3.81, 3.72	m	-	CH_2_	62.3
Sucrose	5.41	d	3.8	CH	93.5
	4.16	d	8.6	CH	78.4
Organic acids					
Succinic acid	2.52	s	-	CH_2_	26.6
Shikimic acid	6.49	m	-	CH(=)	131.8
	-			C	137.3
	4.35	m	-	CH	67.8
	3.95	m	-	CH	68.3
	3.61	dd	4.4; 8.8	CH	73.8
	2.78, 2.19	m	-	CH_2_	34.1
Formic acid	8.48	s	-	CH	169.1
Other					
Choline	3.21	s	-	CH_3_	54.9

* TSP signal (δ^1^H = 0.00 ppm) and methanol (δ^13^C = 49.15 ppm) are used as references. ** Signal multiplicity; s—single; d—doublet; dd—doublet of doublets; t—triplet; m—multiplet; *** n.d.—not detected.

**Table 2 ijms-24-14986-t002:** Metabolite concentration mean in needles collected from the 4 regions exposed to different levels of pollution.

	Formic Acid	Shikimic Acid	Sucrose	α-Glucose	β-Glucose	Fructose	Choline	GABA	Succinic Acid	Alanine	Threonine
Region 4	0.820 b	33.834 ab	1.326 a	13.676 a	21.588 a	6.940 a	3.171 a	2.157 a	12.732 a	2.430 a	1.327 a
Region 3	0.888 b	26.460 b	1.611 a	13.069 a	20.670 a	12.674 a	3.786 a	1.619 a	16.255 a	1.739 a	1.444 a
Region 1	2.591 a	35.932 a	0.586 a	11.551 a	18.493 a	8.247 a	5.643 a	1.948 a	12.308 a	1.860 a	0.902 a
Region 2	0.956 b	38.954 a	0.365 a	13.010 a	20.425 a	7.476 a	3.986 a	1.671 a	10.248 a	1.919 a	1.220 a
Pr > F	<0.0001	0.036	0.207	0.262	0.279	0.147	0.225	0.792	0.690	0.746	0.745
Significant	Yes	Yes	No	No	No	No	No	No	No	No	No

Different letters in each row of the same variant are significantly different at the 0.05 level according to ANOVA by Tukey’s test. Statistical analysis was conducted using one-way ANOVA with pairwise post hoc comparisons and Tukey’s test. The color-coding system enhances the interpretability of the data and provides a clear visual indicator of metabolites with low (blue color) to high (red color) mean variation.

**Table 3 ijms-24-14986-t003:** Geographic origin and habitat characterization of selected conifers.

Site	Code	Geographical Coordinates	Altitude (m)	Species	T_max_ (°C)	T_min_ (°C)	P_an_ (mm)	Pollution Level/Sampling Area Type
Mihaesti	S_1	45.043233, 24.248252	236	Spruce	16.8	7.0	65.5	Low-medium, rural, Region 2
Govora	S_2	45.072151, 24.205248	270	Spruce	16.3	6.8	65.5	Low-medium, rural, Region 2
Govora	S_3	45.072818, 24.195674	282	Fir	16.3	6.8	65.5	Low-medium, rural, Region 2
Govora	S_4	45.075123, 24.192590	299	Fir	16.3	6.8	65.5	Low-medium, rural, Region 2
Baile Govora	S_5	45.081617, 24.177123	329	Spruce	16.0	6.2	65.5	Low-medium, balneo resort, Region 2
Baile Govora	S_6	45.082184, 24.168034	392	Spruce	16.0	6.2	65.5	Low-medium, balneo resort, Region 2
Baile Govora	S_7	45.078622, 24.184012	307	Spruce	16.0	6.2	65.5	Low-medium, balneo resort, Region 2
Baile Govora	S_8	45.076675, 24.188106	300	Spruce	16.0	6.2	65.5	Low-medium, balneo resort, Region 2
Govora	S_9	45.086692, 24.218175	272	Fir	16.3	6.8	65.5	Medium, rural, Region 3
Ocnele Mari	S_10	45.08831, 24.29659	266	Spruce	16.3	6.6	65.5	Low-Medium, rural, Region 2
Ocnele Mari	S_11	45.086189, 24.302747	264	Spruce	16.3	6.6	65.5	Medium, rural, Region 3
Ocnele Mari	S_12	45.081915, 24.309449	259	Spruce	16.3	6.6	65.5	Medium, rural, Region 3
Ocnele Mari	S_13	45.078993, 24.311667	250	Spruce	16.3	6.6	65.5	Medium, rural, Region 3
Troian	S_14	45.072444, 24.330117	246	Spruce	16.8	7.0	65.5	Medium, rural, Region 3
Vladesti	S_15	45.119613, 24.305616	292	Spruce	16.3	6.8	65.5	Medium, rural, Region 3
Vladesti	S_16	45.112791, 24.323620	278	Fir	16.3	6.8	65.5	Medium, rural, Region 3
Vladesti	S_17	45.113277, 24.322721	278	Spruce	16.3	6.8	65.5	Medium, rural, Region 3
Vladesti	S_18	45.127225, 24.271221	313	Fir	16.3	6.8	65.5	Medium, rural, Region 3
Pausesti Maglasi	S_19	45.140121, 24.246315	338	Spruce	15.0	7.6	59.2	Medium, rural, Region 3
Olanesti	S_20	45.172527, 24.257951	378	Spruce	15.0	7.6	59.2	Low-medium, rural, Region 2
Baile Olanesti	S_21	45.203455, 24.241029	434	Spruce	14.0	6.6	59.2	Low-medium, balneo resort, Region 2
Baile Olanesti	S_22	45.206098, 24.237576	422	Spruce	14.0	6.6	59.2	Low-medium, balneo resort, Region 2
Pausesti Maglasi	S_23	45.152544, 24.248061	356	Spruce	15.0	7.6	59.2	Medium, rural, Region 3
Ramnicu Valcea	S_24	45.106805, 24.363972	253	Spruce	16.8	7	65.5	High, urban, Region 4
Ramnicu Valcea	S_25	45.109167, 24.363379	260	Spruce	16.8	7	65.5	High, urban, Region 4
Raureni	S_26	45.03541, 24.28569	220	Spruce	16.8	7.1	65.5	High, industrial, Region 4
Raureni	S_27	45.03541, 24.28569	220	Spruce	16.8	7.1	65.5	High, industrial, Region 4
Cozia National Park	S_28	45.29020, 24.41727	654	Spruce	7.8	0.4	59.2	Low, mountain, Region 1
Cozia National Park	S_29	45.29296, 24.41088	723	Spruce	7.8	0.4	59.2	Low, mountain, Region 1
Cozia National Park	S_30	45.29834, 24.40141	850	Fir	7.8	0.4	59.2	Low, mountain, Region 1
Cozia National Park	S_31	45.30348, 24.39652	855	Spruce	7.8	0.4	59.2	Low, mountain, Region 1
Cozia National Park	S_32	45.30817, 24.38673	907	Fir	7.8	0.4	59.2	Low, mountain, Region 1
Cozia National Park	S_33	45.31366, 24.37694	950	Spruce	7.8	0.4	59.2	Low, mountain, Region 1
Cozia National Park	S_34	45.31998, 24.37551	1036	Spruce	7.8	0.4	59.2	Low, mountain, Region 1
Cozia National Park	S_35	45.32211, 24.37143	1110	Spruce	7.8	0.4	59.2	Low, mountain, Region 1
Cozia National Park	S_36	45.32744, 24.37021	1160	Spruce	7.8	0.4	59.2	Low, mountain, Region 1
Cozia National Park	S_37	45.32851, 24.36409	1180	Fir	7.8	0.4	59.2	Low, mountain, Region 1
Cozia National Park	S_38	45.32898, 24.35904	1311	Fir	7.8	0.4	59.2	Low, mountain, Region 1
Cozia National Park	S_39	45.32631, 24.35373	1310	Spruce	7.8	0.4	59.2	Low, mountain, Region 1
Cozia National Park	S_40	45.32079, 24.33799	1554	Spruce	7.8	0.4	59.2	Low, mountain, Region 1
Cozia National Park	S_41	45.32382, 24.34144	1488	Spruce	7.8	0.4	59.2	Low, mountain, Region 1
Malaia	S_42	45.35751, 24.01593	521	Spruce	14.6	7.3	64.0	Low-medium, rural, Region 2
Voineasa	S_43	45.42373, 23.96654	745	Spruce	12.3	5.3	64.0	Low, mountain resort, Region 1
Voineasa	S_44	45.41644, 23.96439	671	Spruce	12.3	5.3	64.0	Low, mountain resort, Region 1

T_max_—mean annual maximum temperature; T_min_—mean annual minimum temperature; P_an_—mean annual precipitation.

## Data Availability

Not applicable.

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
