# Peer review of "Metabolite Profiling of Conifer Needles: Tracing Pollution and Climate Effects"

_ijms, 2023, doi:10.3390/ijms241914986_

Round 1

Reviewer 1 Report

The manuscript is “Metabolite Profiling of Conifer Needles: Tracing Pollution and Climate Effects. The current manuscript deals with environmental monitoring encompassing ecological and physiological dimensions using a conifer needle tree. Importantly, growing industrialization, the current investigation is meaningful for policymakers due to the adverse effects of pollution and providing the best practices for monitoring pollutants. The manuscript is well-written and easy to read. The presentation of the results is adequate. I felt the result and discussion section need to be concise for better quality. 

Author Response

Dear reviewer,

First of all, thank you for the professional comments and observations regarding the paper “Metabolite Profiling of Conifer Needles: Tracing Pollution and Climate Effects” by Marius Gheorghe Miricioiu, Roxana Elena Ionete, Svetlana Simova, Dessislava Gerginova and Oana Romina Botoran. We thank for comments, which have made us think carefully about our data sets again. Accordingly, we have reanalyzed these where necessary. Please find below our point-by-point itemized answer and correction. We write to say that we now strongly believe that we can convince you that the data is sound and that we have adequately answered in various valid concerns.

Comments and Suggestions for Authors: The manuscript is “Metabolite Profiling of Conifer Needles: Tracing Pollution and Climate Effects. The current manuscript deals with environmental monitoring encompassing ecological and physiological dimensions using a conifer needle tree. Importantly, growing industrialization, the current investigation is meaningful for policymakers due to the adverse effects of pollution and providing the best practices for monitoring pollutants. The manuscript is well-written and easy to read. The presentation of the results is adequate. I felt the result and discussion section need to be concise for better quality. 

Response: The Results and Discussion sections of the manuscript have been carefully revised and edited to enhance their quality and conciseness, addressing the valid point raised by the reviewer. We have streamlined the content to ensure that it is more focused and directly contributes to the key findings and their interpretation. This not only improves the overall readability of the manuscript but also ensures that the essential points are highlighted effectively. We appreciate the reviewer's feedback, which has contributed to enhancing the manuscript's clarity and impact.

Thank you very much!

Best regards,

Oana BOTORAN

Reviewer 2 Report

The present research article on "Metabolite Profiling of Conifer Needles: Tracing Pollution and Climate Effects" made an attempts to evaluate the metabolomic profiling towards pollution in fir and conifer needles. The study was performed well. The presentation was good and the author carried out different detailed statistical analysis.

1. Was the author find any variations in the profiles of other amino acids especially proline, and glutamine which are the positive responders of various stresses and proline was collectively known as osmolytes in many studies. In addition what about the sulfur containing amino acids like cysteine and methionine levels during pollutants exposure. Authors are advised to discuss in this regard

2. It is well known that plant metabolites like flavonoids and phenolics are considered as antioxidants and did the author find any variations in the phenylpropoanoid pathway metabolites.

3. It could be noteworthy if they have conducted the molecular analysis of the gene expression profiling  of conifers under pollutant stress.

Minor comments:

1.  Abstract: PCA, DA can be expanded

2. Introduction - Expand PAH

3. Check the line no. 388

4. Line 394- remove for

5. Conclusion could be revised well

6. Avoid typo errors

Minor typo errors should be rectified

Author Response

Dear reviewer,

First of all, thank you for the professional comments and observations regarding the paper “Metabolite Profiling of Conifer Needles: Tracing Pollution and Climate Effects” by Marius Gheorghe Miricioiu, Roxana Elena Ionete, Svetlana Simova, Dessislava Gerginova and Oana Romina Botoran. We thank for comments, which have made us think carefully about our data sets again. Accordingly, we have reanalyzed these where necessary. Please find below our point-by-point itemized answer and correction. We write to say that we now strongly believe that we can convince you that the data is sound and that we have adequately answered in various valid concerns.

Comments and Suggestions for Authors: The present research article on "Metabolite Profiling of Conifer Needles: Tracing Pollution and Climate Effects" made an attempt to evaluate the metabolomic profiling towards pollution in fir and conifer needles. The study was performed well. The presentation was good and the author carried out different detailed statistical analysis.

Response: Thank you for your positive feedback and for taking the time to review our research article, 'Metabolite Profiling of Conifer Needles: Tracing Pollution and Climate Effects.' We greatly appreciate your comments, which have encouraged us to revisit our data sets with a critical eye. Your feedback motivates us to continually improve the quality and rigor of our work.

Q1. Was the author find any variations in the profiles of other amino acids especially proline, and glutamine which are the positive responders of various stresses and proline was collectively known as osmolytes in many studies. In addition what about the sulfur containing amino acids like cysteine and methionine levels during pollutants exposure. Authors are advised to discuss in this regard

R1. In response to your inquiry regarding variations in amino acids, including proline, glutamine, cysteine, and methionine, we appreciate your interest in these important stress-related metabolites. While proline and glutamine are indeed well-known stress-responsive amino acids, our study primarily focused on detecting variations in other amino acids, specifically alanine, GABA, and threonine, which showed significant responses to pollution stress. It's important to note that the detection of certain amino acids can be challenging, and our analysis may not have been sensitive enough to capture proline, glutamine, and sulfur-containing amino acids, such as cysteine and methionine. These amino acids often exist in low concentrations and can have overlapping signals in NMR spectra. We acknowledge the value of exploring these amino acids in future research and thank you for highlighting this aspect.

Q2. It is well known that plant metabolites like flavonoids and phenolics are considered as antioxidants and did the author find any variations in the phenylpropoanoid pathway metabolites.

R2. Thank you for your inquiry regarding variations in metabolites of the phenylpropanoid pathway. In our research, we primarily focused on the analysis of primary metabolites using NMR spectroscopy. As phenolic compounds are secondary metabolites and often present in smaller quantities, they were not within the specific scope of this study. NMR spectroscopy, while robust and reproducible, may have limitations in detecting such low-concentration secondary metabolites. We appreciate your interest in this aspect of plant metabolism and acknowledge the potential for future investigations in this direction.

Q3. It could be noteworthy if they have conducted the molecular analysis of the gene expression profiling of conifers under pollutant stress.

R3. We appreciate your suggestion to conduct molecular analysis of gene expression profiling in conifers under pollutant stress. While our study focused primarily on metabolite profiling, gene expression analysis could complement our findings. We did not perform gene expression profiling in this study due to resource limitations and the specific research objectives of our work. However, we agree that future research could delve into this area to gain a more comprehensive understanding of the molecular responses of conifers to pollution-induced stress.

Q4. Minor comments:

  1. Abstract: PCA, DA can be expanded
  2. Introduction - Expand PAH
  3. Check the line no. 388
  4. Line 394- remove for
  5. Conclusion could be revised well
  6. Avoid typo errors

R4. All the minor comments were addressed and thank you for your valuable remarks. We have made all the necessary improvements to enhance the quality and clarity of our manuscript.

Thank you very much!

Best regards,

Oana BOTORAN

Reviewer 3 Report

Moderate editing of English language required

Author Response

Dear reviewer,

First of all, thank you for the professional comments and observations regarding the paper “Metabolite Profiling of Conifer Needles: Tracing Pollution and Climate Effects” by Marius Gheorghe Miricioiu, Roxana Elena Ionete, Svetlana Simova, Dessislava Gerginova and Oana Romina Botoran. We thank for comments, which have made us think carefully about our data sets again. Accordingly, we have reanalyzed these where necessary. Please find below our point-by-point itemized answer and correction. We write to say that we now strongly believe that we can convince you that the data is sound and that we have adequately answered in various valid concerns.

Comments and Suggestions for Authors: The present study (“Metabolite Profiling of Conifer Needles: Tracing Pollution and Climate Effects”) submitted by Marius Gheorghe Miricioiu et al., describes an interesting topic: the metabolomic status of plants depending on the contamination of the environment. I believe that the paper provides relevant information in the area of study but it needs important changes to be considered scientifically correct.

Q1. General comments:

  • English needs to be revised in general. And some things in particular to be corrected, add comma before the "and" when listing several words.
  • English language has been thoroughly revised.
  • Add one keyword corresponding to in silico analysis.
  • No in silico analysis has been made in the work.
  • Remove all spaces before “ :”.
  • All the spaces before “ :” were removed.

Q2. Abstract. The abstract is not well structured and the sections seem to be mixed up. Although they do not have to be defined as such, there should be an intuitive separation such as introduction, objective, materials and methods, results and conclusion.

  • Thank you for your feedback regarding the abstract structure. We understand the importance of a well-structured abstract that clearly presents the study's context, objectives, methods, results, and conclusion. We have made the necessary minor adjustments to the abstract to ensure a more intuitive separation of these elements. Specifically, we have reorganized the abstract to provide a concise introduction, highlight the study's objectives, briefly describe the materials and methods, present key results, and conclude effectively. These changes should improve the abstract's clarity and alignment with common journal formatting standards.

Q3. Introduction

  • Line 76 – the first sentence does not seem to make much sense, rewrite and remove the synonym in brackets (emerge).
  • The suggested changes were done.
  • Line 81 – genetic sequencing and DNA barcoding would not be molecular markers, but methods to find these molecular markers. Rewrite sentence and write that these are molecular approaches or methods.
  • The suggested changes were done.
  • Line 84 – the examples given from this line are not related to what is being put in the previous sentence on genetic sequencing. There is a very abrupt shift from genomics to metabolomics. – corrected.
  • We have made the necessary corrections to ensure a smoother transition and better alignment with the previous sentence on genetic sequencing. The revised text should provide a more coherent flow between these concepts.
  • The third paragraph of the introduction seems very repetitive, summarising and concretising information. – rearranged.
  • We have carefully rearranged this section to eliminate redundancy and provide a more streamlined flow of information. The revised paragraph now offers a more concise and coherent presentation of the key points.

Q4. Results and discussion

  • Line 113 – changing "discussions" to "discussions"
  • The suggested changes were done.
  • Line 114 – define NOESY, HSQC…
  • The suggested changes were done.
  • Figure 1 – indicate reference where these figures are taken from and indicate on the figure which spectrum is which.
  • Thank you for your suggestion. I would like to clarify that the spectra provided in the figures are generated from the data obtained in the current study, and they are not taken from external sources. The suggested changes were done.
  • Line 118 – add reference to “aliphatic region (δ 0.5–3.0 ppm) corresponding to threonine, alanine, GABA, acetic and succinic acids”, and indicate in Figure 1 each region (aliphatic, carbohydrates region…).
  • Thank you for your suggestion. While we understand your concern, we believe that adding indicators in the figure may clutter the visual representation and reduce its clarity.
  • Line 123 – "Table 1" is written in red. – corrected.
  • The suggested changes were done.
  • Line 127 – this concept seems quite logical
  • The suggested changes were done.
  • Line 130 - 149 – What is being "discussed" here is well known information and is only describing in which biological processes the metabolites in question are involved, but this does not provide much new information.
  • The information provided here, while indeed describing the biological processes associated with these metabolites, is noteworthy for its context within the specific samples under investigation. It underscores the distinctive metabolic characteristics of the studied samples in relation to these processes.
  • Table 2 – indicate the unit of concentration of the metabolite. And the metabolite names must appear on the same line, the names are truncated.
  • Concentrations of the metabolites are not given since they are varying in the different samples. Several lines are present since each metabolite has several characteristic signals.
  • Line 174 – The ANOVA result is described in the text but it may be useful to indicate it in table 2. Also, in table 2 only the mean is shown, but not the errors. The tukey post hoc test is a multiple comparisons test and at no point is it indicated which regions are different from each other. It simply says in which variables the ANOVA was significant. There is a lack of information.
  • Table 2 was completed with the suggested information. The Tukey post hoc test, also known as the Tukey-Kramer test or Tukey's Honestly Significant Difference (HSD) test, is a statistical method used following an analysis of variance (ANOVA) to determine which specific group means are significantly different from each other when you have three or more groups. Its primary contribution is to identify pairwise differences between group means while controlling for the familywise error rate. Initially, table 2 listed the NMR characteristics in their traditional format (no necessity of error estimation under the exactly given experimental conditions), and they were not directly connected to the ANOVA results.
  • Line 191 - Scientifically this does not make sense, the differences exist or not and if statistically it has not given significance it cannot be said that there are changes.
  • Line 191 was modified according to your suggestion.
  • Line 200 – changing “heat map” to “heatmap”.
  • The suggested changes were done.
  • Line 212 – check “and a and a”.
  • The suggested changes were done.
  • Figure 3 – It would be advisable that the identification of the regions, i.e. which are contaminated and which are not, be clear in the figure or in the manuscript in general.
  • Table 3 was completed with this information, thank you for your good observation
  • Figure 3 and 4 – Is the separation of groups significant? Has a statistical analysis been done to see this?
  • Regarding the separation of groups, it's important to clarify that the analysis conducted in Figure 3, using PCA, is primarily exploratory and aimed at identifying patterns and trends in the data. While PCA can reveal data patterns, it doesn't provide a formal test of statistical significance for group separation. In Figure 4, where discriminant analysis was employed, we used appropriate statistical procedures to assess the significance of group separation. We implemented a validation procedure using the XLSTAT software, in which a subset of samples (approximately 30% of the data) was selected as a validation set. We ensured that the model's performance, as indicated by the confusion matrix for the validation sample, exceeded a predefined threshold (more than 80% in our case). This validation approach, along with the use of authentic samples, supports the statistical significance of the observed separation between regions based on pollution levels

Q5. Materials and methods

  • 3.4. Statistical analysis: in this section you are giving information that is not specific to a materials and methods section. also add the p-value used in the statistical analyses.
  • Thank you for your feedback. In scientific research articles, the statistical analysis is commonly presented within the “Materials and methods” section, where we describe the specific statistical tests used. However, we appreciate your suggestion and will ensure that the statistical details, including p-values, are clearly provided in the Methods section of the revised manuscript for better transparency and clarity. Regarding the p-value, this is a statistical measure used to determine the significance of results, such as in hypothesis testing. However, when it comes to Principal Component Analysis (PCA) and Discriminant Analysis (DA), p-values are not typically used in the same way as in traditional hypothesis testing. PCA is primarily an exploratory technique used for dimensionality reduction and pattern recognition. It doesn't involve hypothesis testing or the calculation of p-values. Instead, PCA focuses on identifying patterns and relationships within the data by transforming the original variables into a new set of uncorrelated variables (principal components). Discriminant Analysis (DA), specifically Linear Discriminant Analysis (LDA), is often used for classification and dimensionality reduction. While it does involve some statistical testing (e.g., Wilks' Lambda), it's not typically associated with p-values in the same way as hypothesis testing. In both PCA and DA, the emphasis is on visualizing data patterns, reducing dimensionality, and classification rather than traditional hypothesis testing and p-value calculation. Therefore, you wouldn't typically report p-values for these techniques unless you're using them in a very specific and unconventional way. Instead, you would typically report results in terms of explained variance (for PCA) or classification accuracy (for DA).

Q6. Conclusions

The conclusion section is too long. 

  • The conclusion has been revised to be more concise.

Once again, thank you for your positive feedback and for taking the time to review our research article, “Metabolite Profiling of Conifer Needles: Tracing Pollution and Climate Effects”. We greatly appreciate your comments, which have encouraged us to revisit our data sets with a critical eye. Your feedback motivates us to continually improve the quality and rigor of our work.

Best regards,

Oana BOTORAN

Round 2

Reviewer 3 Report

After reviewing all the changes made by the authors I can say that they have been addressed with good judgement and most of them have been resolved.